# CLIPAway: Harmonizing Focused Embeddings for Removing Objects via Diffusion Models

**Yiğit Ekin**[*,†], **Ahmet Burak Yildirim**[†], **Erdem Eren Caglar**[†],
**Aykut Erdem**[‡,*], **Erkut Erdem**[♦], **Aysegul Dundar**[†]
[†] Department of Computer Engineering, Bilkent University, Ankara, Turkey
[‡] Department of Computer Engineering, Koç University, Istanbul, Turkey
[*] KUIS AI Center, Koç University, Istanbul, Turkey
[♦] Department of Computer Engineering, Hacettepe University, Ankara, Turkey
[*] Correspondence: yigit.ekin@bilkent.edu.tr

## Abstract

Advanced image editing techniques, particularly inpainting, are essential for seamlessly removing unwanted elements while preserving visual integrity. Traditional GAN-based methods have achieved notable success, but recent advancements in diffusion models have produced superior results due to their training on large-scale datasets, enabling the generation of remarkably realistic inpainted images. Despite their strengths, diffusion models often struggle with object removal tasks without explicit guidance, leading to unintended hallucinations of the removed object. To address this issue, we introduce CLIPAway, a novel approach leveraging CLIP embeddings to focus on background regions while excluding foreground elements. CLIPAway enhances inpainting accuracy and quality by identifying embeddings that prioritize the background, thus achieving seamless object removal. Unlike other methods that rely on specialized training datasets or costly manual annotations, CLIPAway provides a flexible, plug-and-play solution compatible with various diffusion-based inpainting techniques. Code and models are available via our project website: https://yigitekin.github.io/CLIPAway/.

## 1 Introduction

In today's digital era, the demand for sophisticated image editing techniques has surged, with inpainting emerging as a fundamental method for seamlessly removing unwanted elements from images while maintaining visual coherence. Image inpainting has long been studied in both academia and industry. Traditionally, research has predominantly focused on Generative Adversarial Network (GAN)-based methods [21, 17, 12, 16, 14, 26, 32], which have shown notable success in inpainting tasks. However, recent advancements in diffusion models have attracted considerable interest due to their ability to produce high-quality results [24, 28, 1, 31]. A key factor behind the effectiveness of diffusion models is their extensive training on large-scale datasets. By leveraging comprehensive collections of diverse image data, diffusion models can learn complex patterns and correlations, and intricate details, allowing them to inpaint missing regions with exceptional realism.

Despite their strengths, diffusion-based text-guided image inpainting models [28, 1] often encounter challenges in object removal tasks without explicit guidance. When tasked with object removal without explicit text cues to insert a replacement or with the text of "*background*", these models may inadvertently hallucinate the removed object, substituting it instead of erasing it entirely. This issue contrasts with user expectations, as users typically anticipate the erased portion to be seamlessly filled with the background. For example, when removing a person on a surfboard (Figure 1, second row), a diffusion model might insert another person, given the context that surfboards often have people on

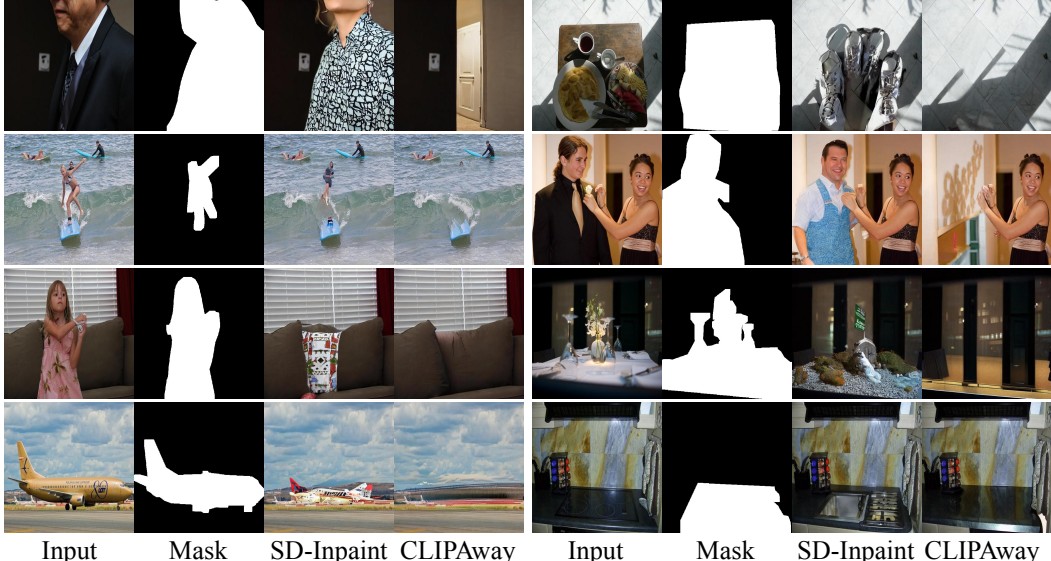

| Input | Mask | SD-Inpaint | CLIPAway | Input | Mask | SD-Inpaint | CLIPAway |

Figure 1: **Diffusion-based inpainting methods often struggle with object removal tasks.** Instead of seamlessly filling the erased area with background elements, diffusion models may unintentionally replace the removed object with another or add irrelevant objects. This outcome diverges from the user's intention, which is typically to restore the area with the background alone, without introducing new elements. Our method, CLIPAway, aims at amending this deficiency by precisely focusing on maintaining the integrity of the background, ensuring that the space is filled as intended by the user.

them. Similarly, when removing a plane from an image of an airport field (Figure 1, fourth row), a model might insert another plane due to the presence of other planes in the background. Alternatively, a model might introduce shoes on the floor (Figure 1, first row) when the user's intention was to fill the space with background elements. In this paper, to address this issue, we propose a novel approach called CLIPAway that leverages AlphaCLIP [25] embeddings to distinguish between foreground (the object to be removed) and background regions. Our method aims to identify an embedding for the inpainted region that focuses on the background while excluding the foreground content, thereby enhancing the quality and accuracy of the inpainting process.

Recent advancements in diffusion-based inpainting for object removal have yielded notable works such as InstInpaint [31], MagicBrush [34], InstructPix2Pix [3], and concurrently ObjectDrop [27]. These methods introduce training datasets with varying approaches: Some generate targets through existing inpainting methods [31], resulting in imperfect targets, while others rely on synthetically generated pairs [3] that may contain annotation errors. Certain methods resort to costly manual annotations [34] or require extensive data collection setups where images of scenes before and after object removal are captured [27]. Furthermore, these techniques typically involve either training a model from scratch or fine-tuning existing models specifically for the removal task. In contrast, our model, CLIPAway, distinguishes itself by its lack of dependency on a specialized training set. It offers a plug-and-play solution compatible with various diffusion-based inpainting methods, ensuring seamless object removal without the need for costly or complex data preparation.

Our contributions can be summarized as follows:

- We introduce CLIPAway, a method that utilizes AlphaCLIP embeddings to effectively differentiate between foreground and background regions for superior object removal.
- Our approach offers a simple plug-and-play solution that does not require specialized training datasets, making it adaptable to various diffusion-based inpainting methods.
- By focusing on background regions, CLIPAway significantly improves the quality and accuracy of inpainting results, avoiding the common issue of object hallucination.
- We provide comprehensive evaluations on a standard dataset, demonstrating consistent improvements over state-of-the-art methods.

## 2 Related Work

Image inpainting involves replacing missing pixels in an image with new ones that blend seamlessly with the surrounding content. Historically, Generative Adversarial Networks (GANs) have dominated this field where they demonstrated significant success across various image domains [21, 17, 33, 12, 16, 32]. However, GAN-based models are typically trained separately for specific image domains, such as face inpainting using training datasets like FFHQ [10] or scenery inpainting using datasets like Places [37]. This domain-specific training restricts their ability to generalize to diverse scenes, thereby limiting their versatility.

Recently, diffusion-based models have made strides, showing promising results [18, 24]. For instance, the Repaint model [18] employs a pretrained unconditional diffusion model to perform image inpainting by conditioning the generation on the unerased parts of the image. Despite its effectiveness, Repaint operates on the image space, thus computationally demanding and slow. Alternative approaches that work in the latent space, e.g. SD-Inpaint [24] and Blended Latent Diffusion [1], adapt the Stable Diffusion (SD) model by adding a mask channel to the latent inputs. These methods, however, often introduce new objects into the scene based on context rather than removing existing ones, conflicting with the user's intention of background restoration. Other diffusion-based methods, such as GLIDE [20] and SmartBrush [28], are designed to add objects rather than remove them.

There has been growing interest in instruction-based inpainting methods for object removal, such as Instruct-Pix2Pix [3] and Inst-Inpaint [31], which use prompts instead of masks for object removal. These methods require datasets specifically tailored for this task. For example, Instruct-Pix2Pix generates paired datasets using the GPT-3 language model [4] and the text-to-image Stable Diffusion model [24], incorporating prompt-to-prompt techniques [6]. While capable of object removal, Instruct-Pix2Pix performs this task with limited precision, possibly due to the synthetic data's lack of diversity or inaccurate annotations. Inst-Inpaint, on the other hand, is trained with paired data where targets are inpainted images generated with GAN-based models. Hence, it inherits the artifacts of these GAN models.

Other works have relied on manual annotation efforts [34], or extensive data collection setups involving scenes captured with and without the object [27]. However, the high cost of manual annotations limits the scale of these datasets. In contrast, our method, CLIPAway, sets itself apart by eliminating the need for specialized training sets. It offers a flexible, plug-and-play solution compatible with various diffusion-based inpainting methods, ensuring seamless object removal without the necessity for costly or complex data preparation.

## 3 Method

### 3.1 Preliminaries

Our framework leverages pretrained diffusion models, particularly the latent diffusion model [24], chosen for its computational efficiency. This model includes an encoder ($E$) and a decoder ($D$). The encoder compresses images into a lower-dimensional latent space while the decoder reconstructs images from these latent codes. These components function similarly to a variational autoencoder and are trained separately from the diffusion process.

The diffusion process, as described by [8], operates on latent codes, denoted as $z_0 = E(x)$, where $x$ is the input image. Noise is gradually added to $z_0$ over a series of time steps $t$ until, after $T$ steps, $z_T$ approximates a normal distribution with zero mean and an identity covariance matrix.

Diffusion models act as denoising autoencoders, trained to reverse the noise addition process. They aim to predict a denoised version of their input, $z_t$, where $z_t$ is a noisy version of $z_0$. The objective function for this denoising task on the latent codes is defined as follows:

$$\mathcal{L}_{LDM} := \mathbb{E}_{E(x),\epsilon \sim N(0,1),t}[\|\epsilon - \epsilon_\theta(z_t, t)\|] \tag{1}$$

Here, $t$ is sampled from the range 1 to $T$, and $\epsilon_\theta(z_t, t)$ represents a neural network, specifically a UNet that predicts the noise added to $z_t$, conditioned on the time step $t$. We specifically employ models fine-tuned for inpainting tasks. These methods involve adding a single-channel mask, which is downsampled to fit the latent space, to the denoising UNet. The diffusion models are commonly trained using text-image pairs, where the text information is extracted from a frozen CLIP text encoder [23]. This encoded text data is then integrated into the UNet via attention layers.

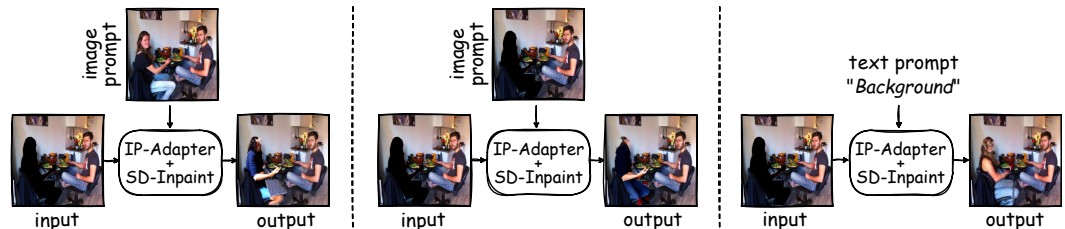

Figure 2: **Limitations of IP-Adapter [30] for Inpainting**. Direct use of the IP-Adapter with the input image as the image prompt is ineffective for inpainting, as it predictably fills the erased area with the original object. In addition, directly giving the prompt "*background*" is also problematic as the background can also contain instances of the images that we want to remove, resulting in a direct replacement of the foreground object. On the other hand, using an erased image as the prompt results in the generation of black artifacts.

## 3.2 CLIPAway

Our objective is to seamlessly remove objects while maintaining the integrity of the background. Unlike conventional inpainting methods that ignore the pixels from the erased area, our approach utilizes these pixels to guide the model on what not to fill in. This distinguishes our method significantly from others. To achieve this, we exploit the detailed pixel-level information available from both the regions to be erased and the unerased regions of the image. While popular Stable-Diffusion models typically rely solely on text conditioning, we explore conditioning the inpainting process on embeddings derived from image pixels.

Recent advancements have introduced additional control signals via adapters, addressing the limitations of text in fully expressing desired outcomes. In some cases, edge maps, poses, or reference images are necessary to effectively control the generation process [35, 36, 19]. The CLIP model utilized for text embedding is originally trained with a contrastive objective jointly with a CLIP image encoder. Adapters have demonstrated that rather than training an image encoder from scratch for reference image-based control, the existing CLIP image encoder can be utilized. UniControl [36] and T2IAdapter [19] extract features from the CLIP image encoder, map them to new features via a trainable network, and concatenate them with text features. These merged features are then fed into the UNet of the diffusion model to guide the image generation process. IP-Adapter [30] further shows that instead of merging image and text features in the cross-attention layer, the features can pass through a small trainable projection network, which are then fed into the UNet via a decoupled attention layer. Our implementation is based on IP-Adapter but can be used with others as well [36, 19].

Our goal is to achieve inpainting by focusing on the background. However, directly using the IP-Adapter with the input image proves ineffective. Using the entire image as the reference (prompt image), the method predictably fills the erased area with the original object (Figure 2, left pane). Conversely, erasing the input image results in black pixels in the masked area, leading to black artifacts in the filled regions (Figure 2, middle pane). Lastly, providing a text prompt as "background", also does not help in removing the object (Figure 2, right pane). Therefore, we need an embedding that solely focuses on the background. To address this, we explore AlphaCLIP [25], which achieves region focus without altering the original image by incorporating regions of interest through an additional alpha channel input. Although it is initialized with the CLIP [22] model, its training requires a substantial set of region-text paired data. By utilizing the Segment Anything Model (SAM) [11] and multimodal large model BLIP-2 [13] for image captioning, millions of region-text pairs are generated. AlphaCLIP model is pretrained on a mixture of region-text pairs and image-text pairs. Their dataset has not been released; fortunately, our method does not require a specialized training set. Instead, we leverage existing models and techniques to achieve our inpainting objectives.

We train a Multi-Layer Perceptron (MLP) model to adapt the publicly released AlphaCLIP image encoder (CLIP-L/14) to the CLIP image encoder used in the IP-Adapter (OpenCLIP ViT-H/14). The MLP consists of six blocks, each containing a linear layer, layer normalization, and GELU activation. It begins with 768 features and outputs 1024 features, matching the output dimensions of the CLIP-L/14 encoder and the OpenCLIP ViT-H/14 encoder, respectively. For this training, we use

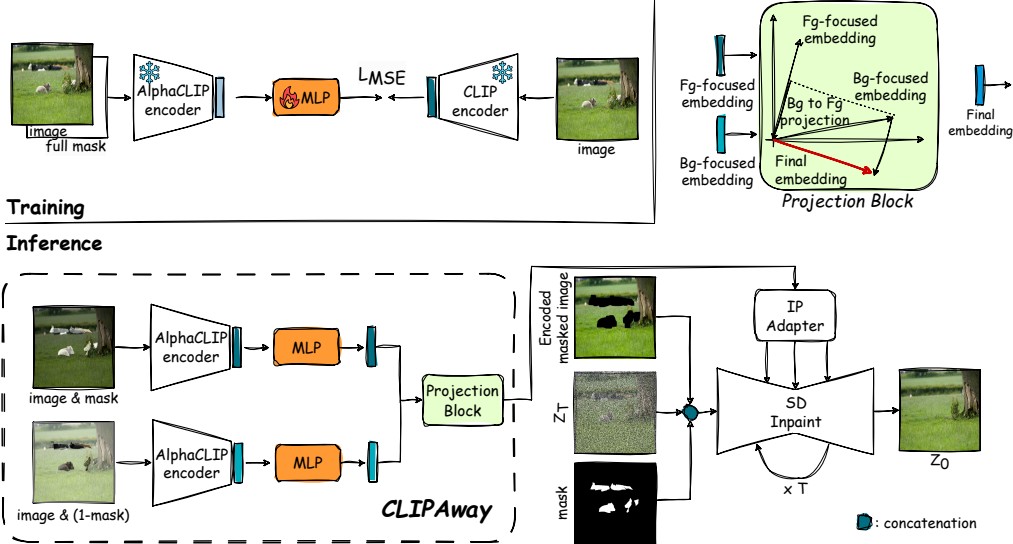

Figure 3: **The overall framework of CLIPAway.** Input images, comprising both foreground and background elements, are embedded via AlphaCLIP. These embedded images are then processed through an MLP trained to adapt features to the IPAdapter input space. Through vector arithmetic on the features, a background embedding without foreground influence is achieved. SDInpaint is depicted as if it is working on the image space for clarity; it works on the latent space.

the COCO image dataset with the alpha channel set to all 1s, corresponding to the full image rather than focusing on a specific region for AlphaCLIP. This setup aligns with AlphaCLIP training, where the authors occasionally set alpha channels to all 1s to indicate full images and sometimes to local regions. In our training, the target is the OpenCLIP embeddings for a given image. This allows us to train a projection layer so that AlphaCLIP outputs features that the rest of the IP-Adapter expects. This part is shown in Figure 3 (Training). We show that AlphaCLIP can be aligned with other CLIP Image encoders without a special dataset.

One of the promises of AlphaCLIP is its ability to focus on a specific region while maintaining contextual awareness. For example, given an image and mask pointing to the background, it may primarily focus on the background while still encoding the foreground, albeit with reduced emphasis. This behavior is illustrated in Figure 4, where we use image prompts as input images with alpha channels corresponding to the mask for foreground focus and the inverse of the mask for background focus. The results shown incorporate our projection layer, which bridges the AlphaCLIP and IP-Adapter. The first row displays the conditional image generations, while the second row shows the inpainting results. When the foreground is focused, the foreground object appears more prominent. Conversely, when the background is focused, the foreground object is present but receives less attention. Therefore, even when the background is focused, the inpaintings still include the object one aims to remove. To remove the foreground overall, we propose to subtract the foreground embedding from the background via projection.

Given two vectors $\mathbf{e_b}$ (background focused embedding) and $\mathbf{e_f}$ (foreground focused embedding), the final embedding $\mathbf{e}_{\text{final}}$ can be calculated as the equation below:

$$\mathbf{e}_{\text{final}} = \mathbf{e_b} - \left( \frac{\mathbf{e_b} \cdot \mathbf{e_f}}{\|\mathbf{e_f}\|} \right) \frac{\mathbf{e_f}}{\|\mathbf{e_f}\|} \tag{2}$$

where $\mathbf{e_b} \cdot \mathbf{e_f}$ is the dot product of $\mathbf{e_b}$ and $\mathbf{e_f}$, and $\|\mathbf{e_f}\|$ is the norm of $\mathbf{e_f}$. With this vector arithmetic, we find the final embedding that is orthogonal to the foreground embedding. After performing this subtraction, the embedding process predominantly focuses on the background, as illustrated in Figure 4 in our results. This tendency is evident in conditional image generation, where the resulting image predominantly exhibits the background style. Consequently, this translates into consistent background filling in the inpainting task for erased areas.

*Conditional Image **Generation** Results*

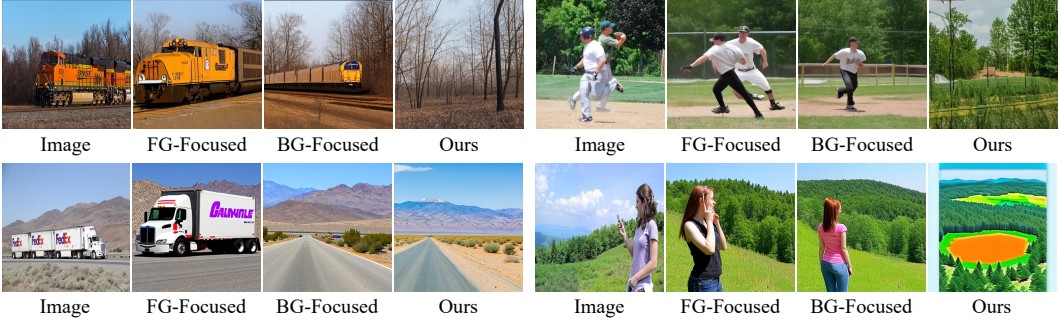

| Image | FG-Focused | BG-Focused | Ours | Image | FG-Focused | BG-Focused | Ours |

| Image | FG-Focused | BG-Focused | Ours | Image | FG-Focused | BG-Focused | Ours |

*Conditional Image **Inpainting** Results*

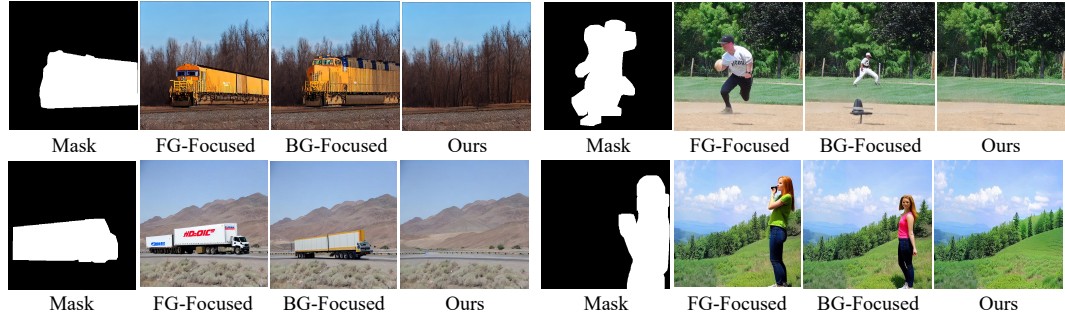

| Mask | FG-Focused | BG-Focused | Ours | Mask | FG-Focused | BG-Focused | Ours |

| Mask | FG-Focused | BG-Focused | Ours | Mask | FG-Focused | BG-Focused | Ours |

Figure 4: Starting with an input image and mask, we present our findings utilizing both foreground and background-focused embeddings. The images in the first row depict the conditional image generation outcomes of the stable-diffusion model without the inpainting task. These visuals offer insights into the focus of the embeddings. While both embeddings capture features from various parts of the image, the foreground embedding tends to emphasize the foreground, whereas the background embedding predominantly focuses on the background but still contains the foreground. Our approach successfully removes the foreground in the generated results, yielding pure background. This outcome is consistent with the image inpainting outputs, as demonstrated in the second row.

The overall framework is depicted in Figure 3 (Inference). Input images containing both foreground and background elements are embedded via AlphaCLIP. These embedded images are then fed into the MLP, which we trained to adapt the features to the IP-Adapter input space. By performing vector arithmetic on the features, we achieve a background embedding without the influence of the foreground. Notably, this method does not necessitate an object removal dataset and can be readily utilized as a plug-and-play feature.

## 4 Experiments

### 4.1 Baselines

We compare our method with state-of-the-art GAN-based and diffusion-based inpainting methods. The GAN based methods include ZITS++ [5], MAT [14], and LaMa [26] models, whereas diffusion based models include Blended Latent Diffusion [1], Unipaint [29], SD-Inpaint [24]. We use the models released by the authors that achieve the best scores.

For GAN-based models, we use the models that are trained on the Places2 dataset. For the diffusion models, we provide an empty prompt. We also experiment with providing prompts as "*background*", but that does not change the results. In our experiments with Unipaint combined with CLIPAway, the masking mechanism proposed in Unipaint is applied to the projected embeddings of our network and then fed into the UNet with the help of IP-Adapter. To demonstrate the flexibility of our method across different CLIP and AlphaCLIP embedding spaces, we also present comparative analyses of our approach when the projection step is applied both before and after the MLP layer. This comparison

employs three distinct AlphaCLIP backbones—ViT-L/14, ViT-L/14@336px, and ViT-B/16—by training an MLP layer tailored to each evaluated backbone.

## 4.2 Datasets

We evaluate the models on the COCO 2017 validation dataset [15], which provides us with the collection of images indoor and outdoor and instance level annotations. For each image in the validation set, we set the masks that correspond to object instances, excluding stuff categories. We dilate the masks with a kernel size of 5 as sometimes the pixels from an object remain in the image and result in artifacts for all models.

## 4.3 Metrics

We report the Frechet Inception Distance (FID) [7], Kernel Inception Distance (KID) [2] and the CLIP Maximum Mean Discrepancy (CMMD) metrics [9] to assess the photorealism of the generated images by comparing the source image distribution with the inpainted image distributions. However, these metrics do not evaluate if the object is correctly removed.

To measure the accuracy of correct object removal, we use the CLIP metrics proposed by Inst-Inpaint [31], namely CLIP Distance and CLIP Accuracy. The goal of CLIP Distance is to evaluate how well the target object is removed. We extract the image regions indicated by bounding boxes from both the source and the inpainted images, then estimate the CLIP similarities [22] between these regions. We expect a larger distance if the object is correctly removed. For CLIP Accuracy, we utilize CLIP as a zero-shot classifier. We identify the most likely semantic label of the image region extracted from the source image using the object's bounding box, considering the object classes in our dataset. Next, we perform another prediction for the image region extracted from the inpainted image. We expect the class prediction to change after performing the object removal operation. If the predicted class based on the source image is not in the Top-1, Top-3, or Top-5 predictions of the inpainted image, it is considered a success. We report the percentage of successes.

We also conduct a user study with 20 participants on the first 20 samples of the validation set. We include ZITS++, Unipaint, SD-Inpaint, and CLIPAway to provide a range of the best-performing models. Participants are asked to evaluate whether the object is correctly removed and to assess the quality of the inpainting, and then select the best result among all choices. Details of the study are given in the Appendix.

## 4.4 Results

We provide quantitative and qualitative comparisons of our method with state-of-the-art inpainting models in Table 1 and Figure 6, respectively. Our approach, when combined with various diffusion-based inpainting methods, demonstrates consistent improvements in both FID and CLIP metrics. Figure 5 illustrates that CLIPAway (with SD-Inpaint) achieves significantly better CLIP@3 and FID scores, positioning it in the top left corner, indicating more accurate and higher quality results compared to other methods. Additionally, our user study shows a strong preference for our approach over competing methods. Human subjects preferred our method 71.75% of the time, compared to 13.25% for ZITS++, 11.25% for SD-Inpaint, and 3.75% for Unipaint.

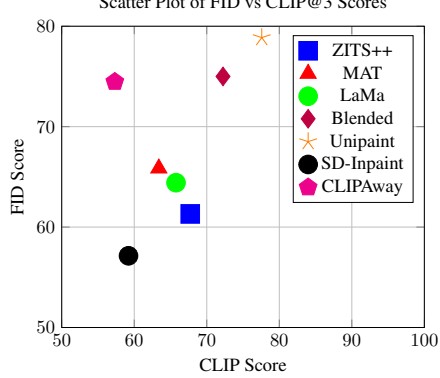

Figure 5: Comparison of CLIPAway with state-of-the-art methods based on image quality and inpainting accuracy.

A few predictions from competing models are presented in Figure 6. GAN-based models, namely LaMa, MAT, and ZITS++, do not exhibit issues with object insertion. However, this problem is evident in the outputs of diffusion-based methods. This may be because diffusion models are more powerful and better at modeling data distribution, leading them to generate objects that more closely match the real distribution. Although GAN-based models avoid object insertion object insertion, they fail to

| Models | FID ↓ | KID ↓ | CMMD ↓ | CLIP Dist.↑ | CLIP@1 ↑ | CLIP@3 ↑ | CLIP@5 ↑ |
|---|---|---|---|---|---|---|---|
| ZITS++ [5] | 67.72 | 0.0208 | 0.74 | 0.66 | 76.15 | 61.31 | 52.91 |
| MAT [14] | 63.39 | 0.0278 | 0.93 | 0.76 | 80.62 | 65.85 | 60.10 |
| LaMa [26] | 65.76 | 0.0195 | 0.81 | 0.66 | 78.34 | 64.42 | 56.85 |
| SD-Inpaint + LaMa | 51.33 | 0.0117 | 0.45 | 0.75 | 72.29 | 57.61 | 50.01 |
| Blended Diff. [1] | 72.24 | 0.0362 | 0.89 | 0.85 | 85.69 | 75.01 | 69.34 |
| + CLIPAway | 61.66 (-10.58) | 0.0194 (-0.0168) | 0.78 (-0.11) | 0.83 (-0.02) | 87.28 (+1.59) | 78.87 (+3.86) | 73.20 (+3.86) |
| Unipaint [29] | 77.58 | 0.0360 | 0.98 | 0.78 | 85.38 | 74.48 | 67.44 |
| + CLIPAway | 62.18 (-15.40) | 0.0199 (-0.0161) | 0.79 (-0.19) | 0.84 (+0.06) | 88.26 (+2.88) | 78.65 (+4.17) | 73.05 (+5.61) |
| SD-Inpaint [24] | 59.21 | 0.0145 | 0.54 | 0.75 | 70.45 | 57.14 | 49.88 |
| + CLIPAway | 57.32 (-1.89) | 0.0108 (-0.0037) | 0.53 (-0.01) | 0.81 (+0.06) | 84.82 (+14.37) | 74.42 (+17.28) | 67.76 (+17.88) |

Table 1: **Evaluation results.** Improvements of CLIPAway over the base models are given in parenthesis for each metric.

generate realistic backgrounds. Our method, CLIPAway, is the only one that effectively removes objects and fills the regions realistically. For example, in the first row, GAN-based models fill the inpainted area in a blurry way, and diffusion models insert a person. In contrast, our method removes the person and fills the area seamlessly. In the third example, our approach is the only one that realistically fills the kitchen background. Similarly, in the eighth example, while GAN-based models extend the object in a blurry way and diffusion models add unrelated content, our method provides a sharp and realistic result.

Table 2 illustrates the adaptability of our approach across various CLIP and AlphaCLIP embedding spaces. Our method is not constrained to the CLIP embedding spaces employed in our initial experiments; with different backbones, the results consistently enhance the performance of the SD-Inpaint model across diverse CLIP embeddings. Additionally, our projection method is applicable beyond the OpenCLIP embedding space. Since AlphaCLIP's vision transformer is trained with objectives similar to those of the CLIP vision transformer, the resulting feature spaces are conceptually aligned. Consequently, projections can be effectively performed in the AlphaCLIP embedding space or in other CLIP embedding spaces with comparable properties. To validate this, we evaluated the projection method on the AlphaCLIP feature space (projection on AlphaCLIP space followed by MLP) using the same experimental setup.

| Models | FID ↓ | CMMD ↓ | CLIP Dist.↑ | CLIP@1 ↑ | CLIP@3 ↑ | CLIP@5 ↑ |
|---|---|---|---|---|---|---|
| SD-Inpaint [24] | 59.21 | 0.54 | 0.75 | 70.45 | 57.14 | 49.88 |
| *projection before MLP* | | | | | | |
| SD-Inpaint + CLIPAway (VIT L/14) [24] | 57.32 | 0.53 | 0.81 | 84.82 | 74.42 | 67.76 |
| SD-Inpaint + CLIPAway (VIT L/14@336px) [24] | 54.93 | 0.48 | 0.80 | 82.36 | 71.68 | 63.28 |
| SD-Inpaint + CLIPAway (ViT-B/16) [24] | 55.31 | 0.48 | 0.78 | 83.57 | 72.44 | 63.81 |
| *projection after MLP* | | | | | | |
| SD-Inpaint + CLIPAway (VIT L/14) [24] | 56.15 | 0.42 | 0.86 | 85.31 | 74.26 | 68.58 |
| SD-Inpaint + CLIPAway (VIT L/14@336px) [24] | 54.46 | 0.36 | 0.82 | 82.13 | 70.02 | 63.28 |
| SD-Inpaint + CLIPAway (ViT-B/16) [24] | 54.99 | 0.41 | 0.84 | 85.08 | 74.79 | 68.35 |

Table 2: **Evaluation results.** SD-Inpaint and SD-Inpaint + CLIPAway across different CLIP embedding configurations, with projections evaluated before and after MLP.

Figure 7 shows our method integrated with various diffusion-based inpainting techniques, highlighting the significant performance enhancement our module offers. In Figure 8, we present qualitative comparisons between our method and instruction-based diffusion models. Since our method requires a mask and these models require prompts, this is not a direct one-to-one comparison. The visual results reveal that Instruct-Pix2Pix [3] struggles to accurately remove an object, and Inst-Inpaint [31] produces blurry inpainted images due to its training with GAN-generated targets. It is important to note that these competing methods rely on specialized datasets and specific model training for this task. In contrast, our method does not necessitate a special dataset and can seamlessly integrate with existing diffusion models.

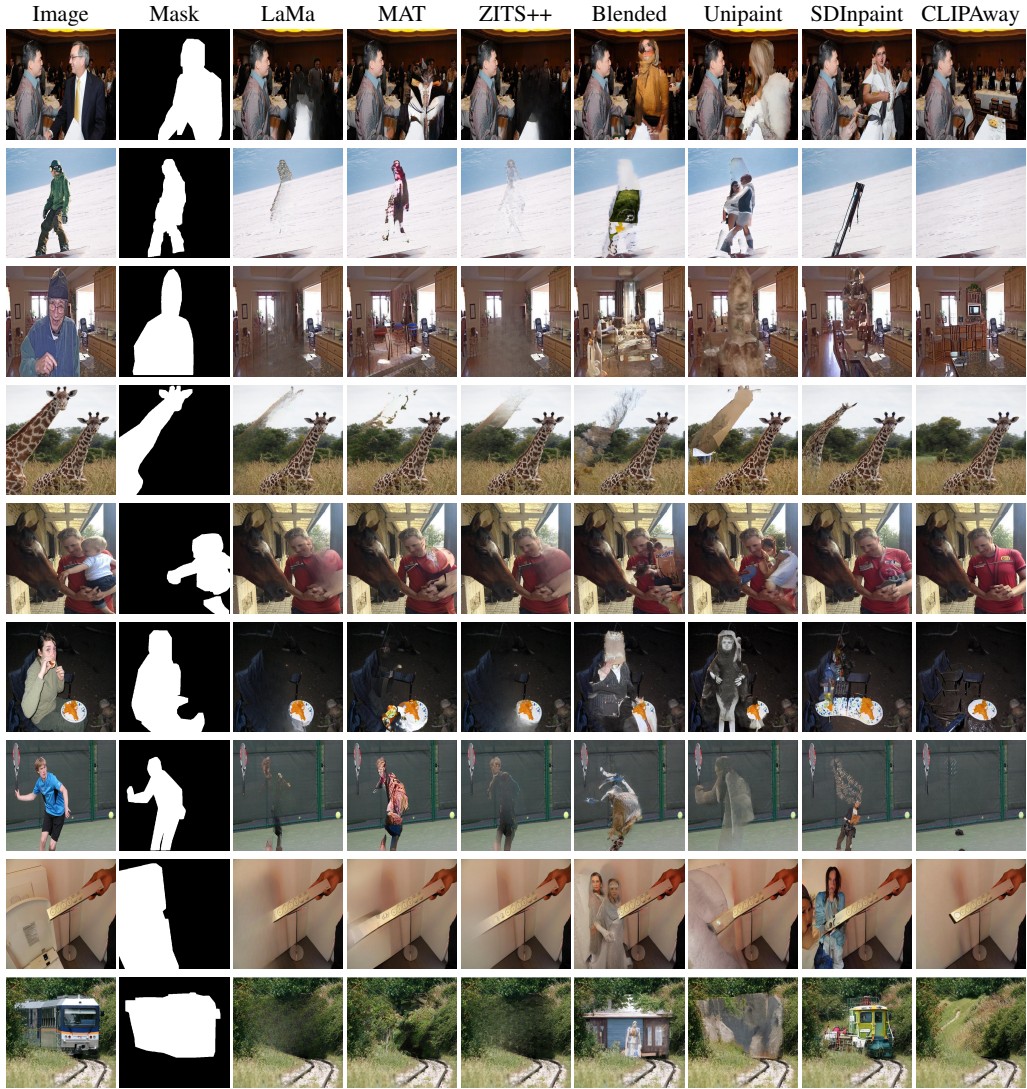

Figure 6: Diffusion models often replace the removed object or insert new content instead of simply removing it. GAN-based models avoid adding new objects but struggle to generate realistic backgrounds. Our method, CLIPAway, effectively removes objects and fills the regions with realistic background content.

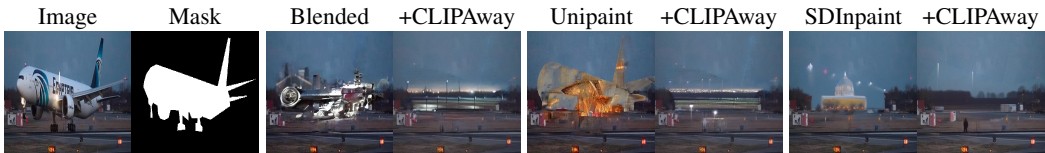

Figure 7: Qualitative results of diffusion-based models and our method combined with them.

## 5 Conclusion and Broader Impacts

In this paper, we introduced CLIPAway, a novel approach for object removal in images using diffusion-based inpainting methods. Specifically, CLIPAway addresses the common issue of diffusion models hallucinating removed objects by focusing embeddings on the background. Our method leverages AlphaCLIP embeddings to effectively distinguish between foreground and background regions, focusing on background restoration to achieve seamless and realistic inpainting. By eliminating

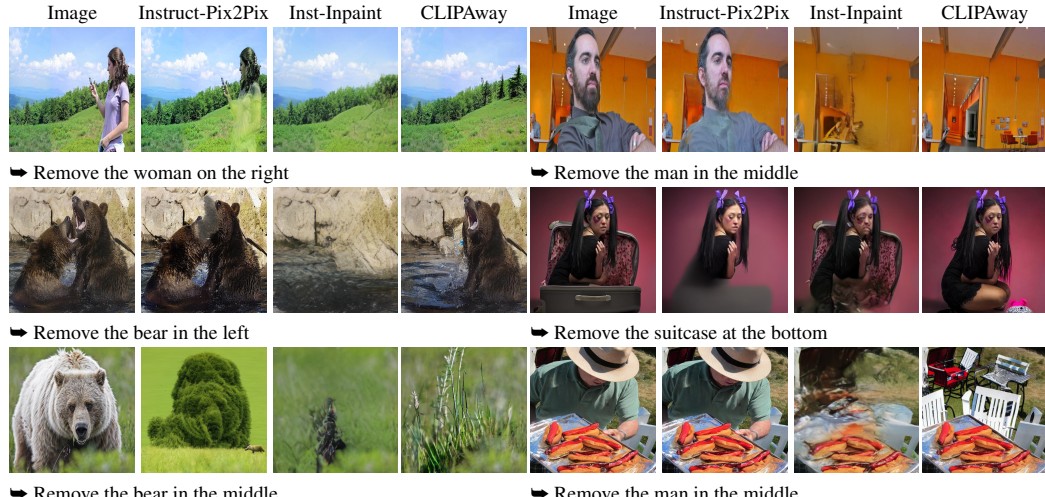

| Image | Instruct-Pix2Pix | Inst-Inpaint | CLIPAway | Image | Instruct-Pix2Pix | Inst-Inpaint | CLIPAway |

➡ Remove the woman on the right        ➡ Remove the man in the middle

➡ Remove the bear in the left        ➡ Remove the suitcase at the bottom

➡ Remove the bear in the middle        ➡ Remove the man in the middle

Figure 8: Qualitative results comparing instruction-based diffusion models with our method, which utilizes a mask for object removal.

the dependency on specialized training datasets, CLIPAway provides a flexible, plug-and-play solution compatible with various diffusion-based inpainting techniques. Our extensive experiments demonstrated that CLIPAway significantly improves the quality and accuracy of inpainting compared to state-of-the-art methods.

**Broader Impacts.** Our framework has the potential to image restoration, editing, and completion. However, this technology also brings important ethical considerations. One potential misuse is in the alteration or falsification of visual content, leading to the creation of misleading or deceptive images. We do not endorse such activities and emphasize the necessity of establishing safeguards to ensure the ethical use of this technology.

**Limitations.** Our method is demonstrated using latent-based diffusion models. A limitation is that, despite the diffusion occurring in the latent space, these models are still slower than GAN-based methods and do not run real-time. Another limitation of our model is the degradation in performance when the preferred resolution is not used. For instance, when utilizing the SD-Inpaint pipeline [1], the expected resolution for inference without quality loss is 512×512. If a different resolution is provided, performance degrades. While this issue is inherent to latent-based diffusion models rather than specific to our approach, it remains a present limitation. Additionally, our model, while removing objects, does not remove their shadows if they are not included in the mask. This can be seen in Figure 1 in the first two examples on the right, where shadows are handled as patterns by the model instead of being removed.

## Acknowledgments

This work was supported in part by KUIS AI Research Award. We thank all the reviewers for their valuable comments.

---

[1] https://huggingface.co/botp/stable-diffusion-v1-5-inpainting

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

# A Supplementary Material

In this supplementary material we provide:

1. Training and inference algorithm
2. Architectural details of the MLP network and training details
3. Conditional image generation examples for projected embeddings
4. Visual comparison with LaMa, MAT, ZITS++, Blended Latent Diffusion, Unipaint and SD-Inpaint
5. Details of the user study that we have conducted
6. Potential extension to SDXL-Inpainting pipeline
7. Our models flexibility across CLIP embedding styles
8. Our models contextual understanding
9. Reference-based background inpainting examples of our model
10. LaMa + SDInpainting comparison with our model

## A.1 Training and Inference Algorithm

---

**Algorithm 1** Algorithm for our training and inference

---

1: **Training Algorithm Of MLP Projection Layer:**
2: **Input:** Training data $D = \{(x_i)\}_{i=1}^n$, Alpha-CLIP Image Encoder ($\alpha$-CLIP), CLIP Image Encoder (CLIP), MLP with parameters $\theta$, learning rate $\eta$, number of epochs $E$
3: **Output:** Model parameters $\theta$
4: **for** epoch = 1 to $E$ **do**
5:    **for** each $(x_i) \in D$ **do**
6:       mask $\leftarrow$ ones($x_i$.shape)
7:       $e_{\alpha-\text{clip}} \leftarrow \alpha$-CLIP(mask,$x_i$)
8:       $e_{\text{projected}} \leftarrow \text{MLP}(e_{\alpha-\text{clip}})$
9:       $e_{\text{clip}} \leftarrow \text{CLIP}(x_i)$
10:      $L_{MSE} \leftarrow ||e_{\text{clip}} - e_{\alpha-\text{clip}}||_2$
11:      $\theta \leftarrow \theta - \eta\nabla_\theta L_{MSE}$
12:    **end for**
13: **end for**
14: **Inference Algorithm of CLIPAway:**
15: **Input:** Image $I_{in}$, Mask $M$, downscaled mask $m$, Alpha-CLIP Image Encoder ($\alpha$-CLIP), Trained MLP from 1.1, trained Stable Diffusion model SD, trained IP-Adapter, a timestep $t$
16: **Output:** Inpainted Image $I_{out}$
17: $Z_T \sim \mathcal{N}(0, I)$
18: $e_{fg} \leftarrow \alpha$-CLIP(mask,$I_{in}$)
19: $e_{bg} \leftarrow \alpha$-CLIP(1 - mask,$I_{in}$)
20: $\hat{e_{fg}} \leftarrow \text{MLP}(e_{fg})$
21: $\hat{e_{bg}} \leftarrow \text{MLP}(e_{bg})$
22: $e_{\text{final}} \leftarrow \hat{e_{bg}} - \left(\frac{b \cdot f}{||f||^2}\right) f$
23: $e_{ip} \leftarrow \text{IP-Adapter}(e_{\text{final}})$
24: $I_{out} \leftarrow SD(Z_T, t, e_{ip})$

---

## A.2 Architectural details of the MLP network and training details

As stated, we have trained a multi-layer perceptron (MLP) to project Alpha-CLIP embeddings into CLIP embeddings subspace. The network consists of 6 blocks each containing a linear layer followed by a layer normalization followed by a GELU activation function. The dimension of linear layers changes midway to adapt to a different embedding dimension, transitioning from Alpha-CLIP embedding dimension to IP-Adapter embedding dimension. It is also important to note that the final layer does not include a GELU activation function. Complete implementation of the model in a Pytorch-like syntax is given in Table 3.

The MLP model is trained on a single NVIDIA A40 GPU with batch size 8. Adam, optimizer is used with learning rate and weight decay are set to $1e^{-5}$ and $1e^{-4}$, respectively. The model is trained for 300k steps on COCO2017 which took approximately 7 GPU hours.

| Layer | Description |
|-------|-------------|
| 1 | Linear(768, 768) |
|   | LayerNorm(768) |
|   | GELU() |
| 2 | Linear(768, 768) |
|   | LayerNorm(768) |
|   | GELU() |
| 3 | Linear(768, 1024) |
|   | LayerNorm(1024) |
|   | GELU() |
| 4 | Linear(1024, 1024) |
|   | LayerNorm(1024) |
|   | GELU() |
| 5 | Linear(1024, 1024) |
|   | LayerNorm(1024) |
|   | GELU() |
| 6 | Linear(1024, 1024) |
|   | LayerNorm(1024) |
|   | GELU() |
| 7 | Linear(1024, 1024) |

Table 3: Layer-by-Layer Description of the MLP

## A.3 Conditional Image Generation Examples

Image      Mask      Fg Focused   Bg Focused   Projected

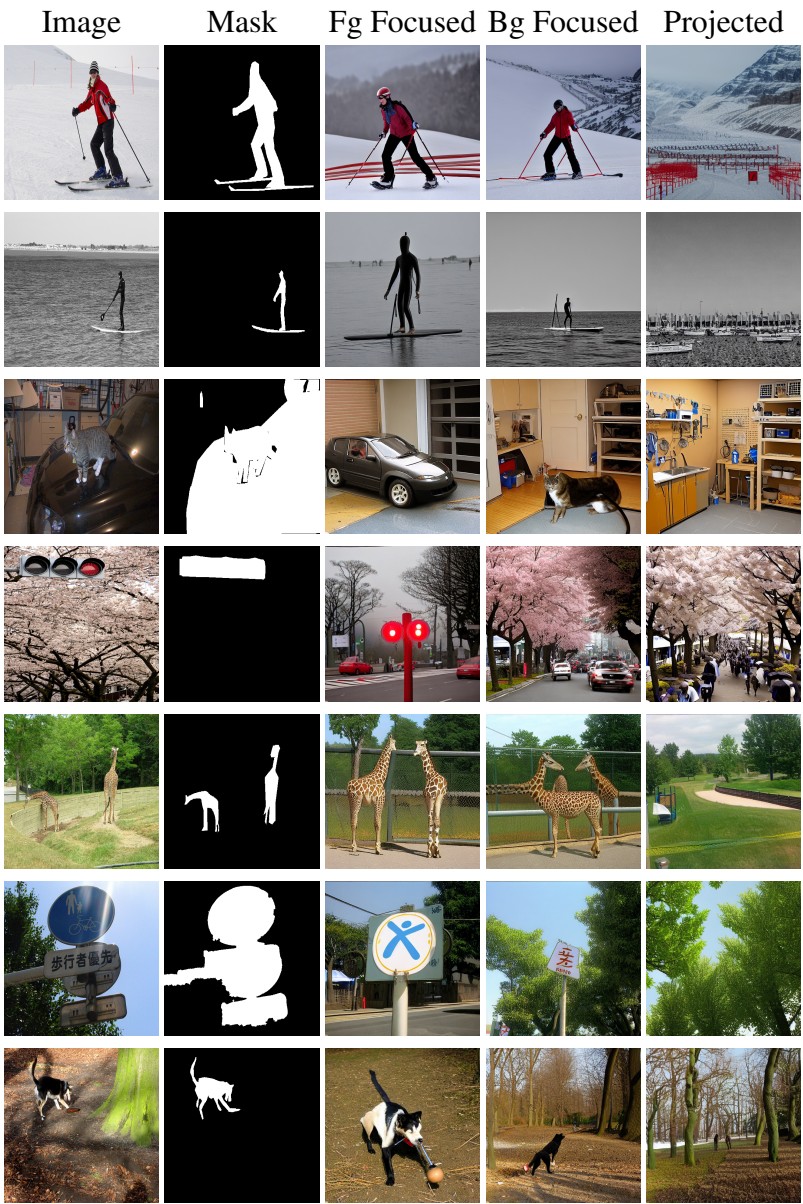

Figure 9: Conditional image generation results (not inpainting), when condition is foreground, background, and projected embeddings.

## A.4 Additional Qualitative Results

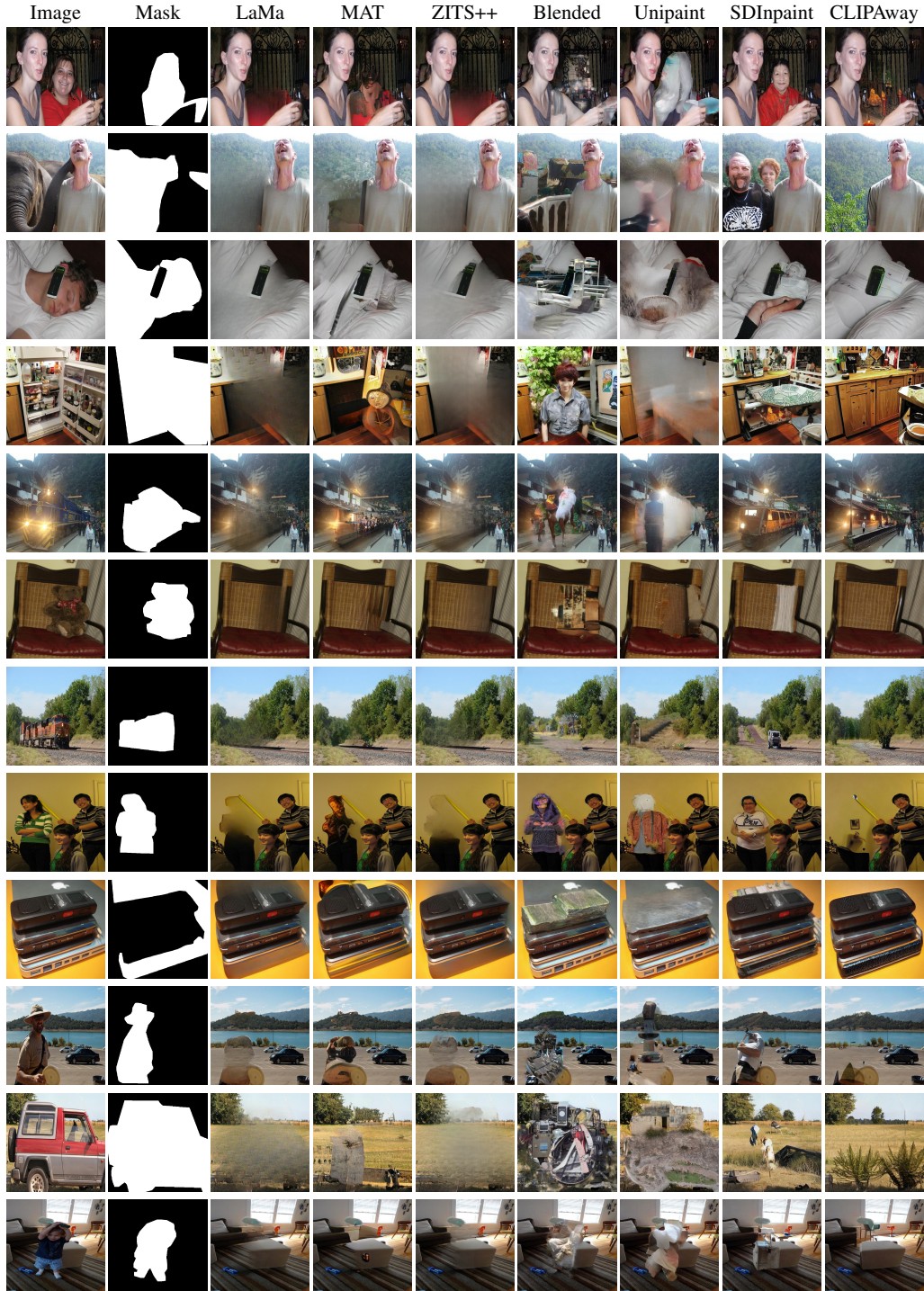

Figure 10: Qualitative results of our and competing methods on the COCO2017 Validation dataset. Diffusion models often replace the object or insert new content instead of removing it, while GAN-based models, although not struggling with object insertion, fail to generate a realistic background. Our method is the only one that effectively removes objects and fills the regions in a realistic manner.

### A.5 User Study Details

We have conducted a user study to obtain a better comparison between our model and our competitors. 20 people voted as volunteers. Random 20 samples are included in our user study without cherry-picking. We include ZITS++, Unipaint, SD-Inpaint, and CLIPAway to have a variety of the best-performing models. The voters were asked to select the option that achieved the best object removal regarding realism and consistency with the scene. The instructions and an example from the study are given in Figure 11 and 12, respectively. The order of the methods are randomized for each question. There was no time limit set in the study.

# User Study - CLIPAway

Image inpainting is a technique used to fill in damaged or missing parts of an image to make it look complete and natural. This process can be used to remove unwanted objects from photos, repair old or damaged photographs, or fill in gaps when parts of an image are missing. The goal is to make the final image appear as natural and seamless as possible, as if the missing or unwanted parts were never there.

**Task:** You will be provided with the original image and a mask indicating the region to be removed in each question. Review the images in options A through D, assessing them based on the following criteria:

- **Accuracy:** If the object is correctly removed.
- **Smoothness:** The texture and transitions in the inpainted area, ensuring they are seamless.

Figure 11: Guideline given to the voters for correctly completing the user study

### A.6 Extension to SDXL-Inpainting Pipeline

SDXL-Inpainting is another widely-used pipeline for inpainting tasks, yet it has notable limitations when applied to object removal. Upon examination, we observed that issues related to unintended object additions are particularly pronounced. SDXL-Inpainting recommends setting the strength parameter to 0.99, using the unerased image as input, and applying noise at this strength level [2]. The strength parameter, which ranges from 0 to 1, controls the degree of transformation applied to the masked area, with higher values introducing more noise and necessitating additional denoising steps. When set to 1, the strength parameter applies maximum noise, often resulting in a degradation of image quality, as noted in the documentation's limitations. Conversely, using a strength of 0.99 frequently causes the model to regenerate erased foreground objects, as residual information about the object remains. Under this configuration, SDXL-Inpainting achieves a CLIP-1 score of 67.22, the lowest among those reported in Table 1 of the main paper. However, when combined with our CLIPAway method, the SDXL-Inpaint model achieves an improved CLIP-1 score of 85.91. These results are presented in Figure 17, which illustrates that at a strength of 1, the model tends to hallucinate or replace objects, while at a strength of 0.99, it may partially regenerate the object.

### A.7 Flexibility Across CLIP Embedding Styles

CLIPAway demonstrates robust adaptability across various CLIP embedding spaces. To explore this flexibility, we trained the MLP for AlphaCLIP with additional model keys—VIT-L/14@336px and VIT-B/16—alongside VIT-L/14. Each model was subsequently evaluated, and the results underscored

---

[2]https://huggingface.co/diffusers/stable-diffusion-xl-1.0-inpainting-0.1

2. In which image the object is best removed? *

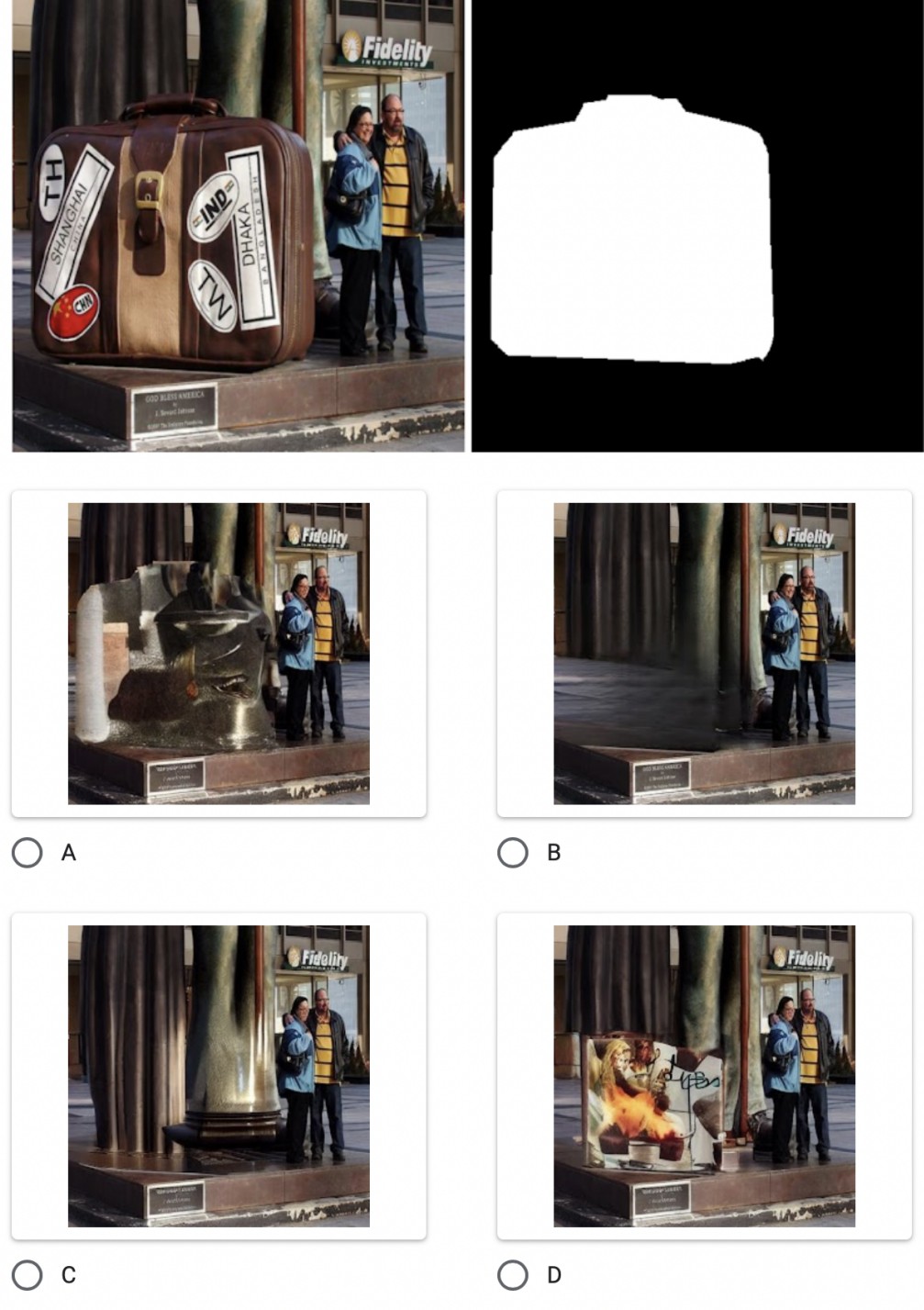

Figure 12: An example question from the user study

the effectiveness of our approach across diverse CLIP embeddings, as shown in 2. This analysis validates that our method can seamlessly operate within different embedding spaces.

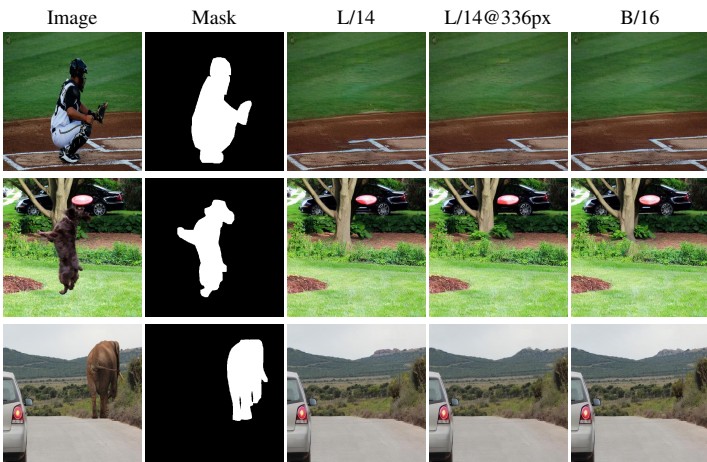

Figure 13: CLIPAway results with different CLIP embedding spaces.

Our projection method is not confined to the OpenCLIP embedding space. As AlphaCLIP's vision transformer and CLIP's vision transformer are trained with aligned objectives, their feature spaces are conceptually similar. Consequently, the projection method can be effectively applied within the AlphaCLIP embedding space or other similar CLIP embedding spaces. To validate this, our projection approach on the AlphaCLIP feature space is evaluated. The outputs of this approach, projection on AlphaCLIP space followed by MLP, confirmed that our method is applicable beyond the OpenCLIP embedding space.

## A.8 Contextual Understanding

As our approach leverages SDInpaint as its backbone, the inpainting process is constrained to regions delineated by the provided masks, a characteristic shared with other competing methods, such as LAMA, MAT, and ZITS++. Additional experiments incorporating both the primary objects and their associated shadow regions within the input masks were conducted. The results, presented in 14, demonstrate that our method successfully eliminates both objects and their corresponding shadows when the mask encompasses both elements.

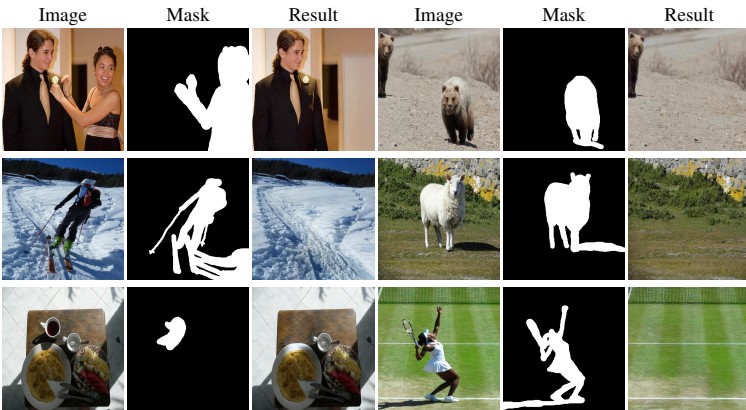

Figure 14: Shadows can be included in the masks if preferred.

## A.9    Reference-based Background Inpainting

While the primary focus of our work centers on object removal, our framework demonstrates versatility across various tasks. We explore the application of our method to reference-based background inpainting, as illustrated in 15. By computing the projected embedding of a reference image that represents the target background, our framework successfully inpaints the background while maintaining the integrity of the foreground. These results highlight the broader potential of our approach in addressing general image manipulation and inpainting challenges.

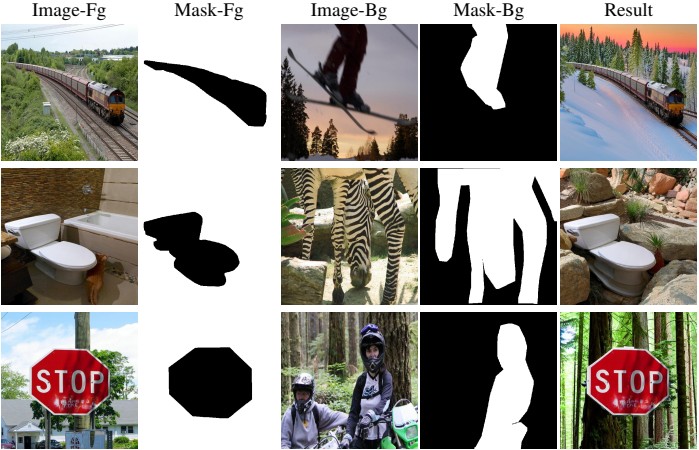

Figure 15: Reference-based background inpainting with CLIPAway.

## A.10    LaMa + SDInpainting

To further evaluate the effectiveness of our approach, we compared our method to a widely-used community pipeline for object removal[3]. This approach first applies LaMa inpainting as a preprocessing step to the SD-Inpaint pipeline, creating a hybrid method intended to capitalize on the strengths of both models. While the integration of SD-Inpaint as a post-processing step enhances overall visual quality metrics, as shown by improved FID scores, it also introduces certain limitations. Specifically, a reduction in object removal effectiveness is noted, primarily due to SD-Inpaint's tendency to hallucinate or introduce extraneous objects in the inpainted regions. Our CLIPAway model notably surpasses this hybrid approach, achieving superior KID (Kernel Inception Distance) metrics and consistently higher scores in CLIP-based object removal metrics. This result is illustrated in Figure 16, where, although LaMa provides a more refined intermediate image for the SD-Inpaint pipeline, the tendency of SD-Inpaint to replace or hallucinate objects remains evident.

---

[3]https://github.com/Mikubill/sd-webui-controlnet/discussions/1597

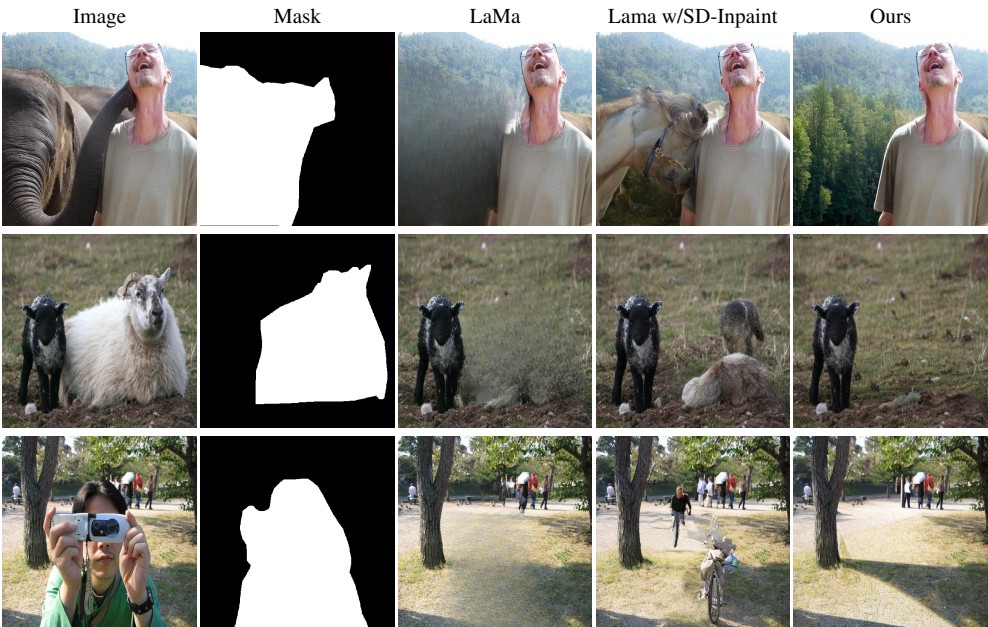

Figure 16: LaMa + SD-Inpaint comparison with Ours.

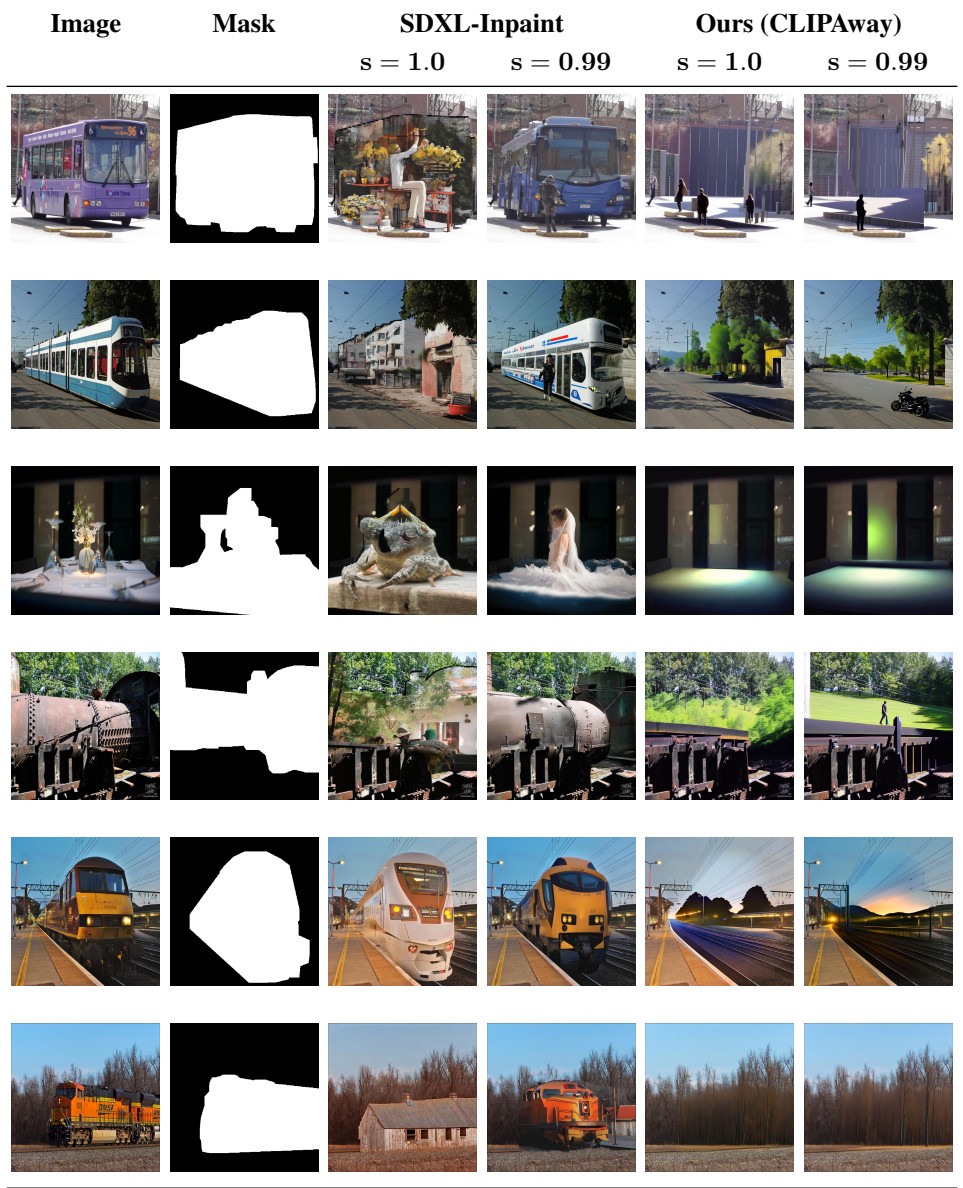

Figure 17: SDXL-Inpaint and SDXL-Inpaint+CLIPAway results with strength values of 1.0 and 0.99.

