# OpenReview forum: "CLIPAway: Harmonizing focused embeddings for removing objects via diffusion models"
_NeurIPS.cc/2024/Conference — NeurIPS 2024 poster_

### Official Review · Reviewer_GcHN · 2024-07-06

**Soundness:** 3
**Presentation:** 3
**Contribution:** 3
**Rating:** 7
**Confidence:** 4

**Summary:**

This work addresses the object removal problem with a simple and effective embedding arithmetic strategy. This idea is implemented by combining the object semantic understanding ability from alpha-clip and the generative ability from the text-to-image diffusion models. One good property is that the proposed method is a plug-and-play strategy that can be adopted in various Diffusion-based t2i frameworks.
Compared to the previous GAN-based and Diffusion-based inpaint methods, this method demonstrates better results with suitable background and fewer artifacts.

**Strengths:**

The simple and effective embedding arithmetic strategy is intuitive and suitable for the object removal problem.

The designed technical framework of the alpha-clip plus t2i diffusion model is flexible and powerful since many powerful conditional image generative models are based on clip embeddings. It would be interesting to see how this technique can be applied to other t2i frameworks.

Figure 5 demonstrates the superiority of the proposed method which performs a good trade-off between FID and CLIP Scores. Figure 6 also shows good inpaint results with fewer artifacts and blurry effects.

The paper is easy to follow and the algorithm has been illustrated step-by-step.

I highly recommend the authors release code and model for the following works. I believe it can benefit the related CV fields.

**Weaknesses:**

It seems that the applicable image resolution is limited by the diffusion method. Comparisons on higher resolutions could demonstrate the boundary of this technique since one benefit LaMa-like methods is the applicability to high-resolution images.

One popular baseline of “LaMa + SDinpaint” should be compared, please check https://github.com/Mikubill/sd-webui-controlnet/discussions/1597

More failure cases could be presented and analyzed to show the limitations.

I suppose this method can be extended to other frameworks like SDXL, it would be great to perform these extensions to make this work more comprehensive.

**Questions:**

“LaMa + SDinpaint” is required to be compared.
Comparisons on higher resolutions could be demonstrated.

**Limitations:**

More failure cases could be discussed.

---

> ### Author Rebuttal · Authors · 2024-08-06
>
> We would like to thank the reviewer for their detailed feedback which help us improve the paper further.
>
> **Applicability to Higher Resolution Images**\
> We appreciate the reviewer's insight regarding the applicability of our method to higher resolution images. As noted, the current limitation is inherent to existing diffusion-based image inpainting methods. Our primary goal was to enhance object-removal performance within these existing constraints. Future research could indeed focus on adapting our method for higher resolutions, potentially bridging the gap between diffusion-based techniques and LaMa-like methods. This would involve significant advancements in diffusion model architectures to handle high-resolution images effectively.
>
> **Comparison with “LaMa + SD-Inpaint” Baseline**\
> Thank you for suggesting the comparison with the “LaMa + SD-Inpaint” baseline. This is  a very interesting work. We conducted this comparison using the same setup as in Table 1 of our manuscript. The results are as follows.
>
> | Method            | FID  | CMMD | CLIP Dist | CLIP@1 | CLIP@3 | CLIP@5 | KID    |
> |-------------------|------|------|-----------|--------|--------|--------|--------|
> | SD-Inpaint              | 59.21| 0.54 | 0.75      | 70.45  | 57.14  | 49.88  | 0.0145 |
> | LaMa              | 65.76| 0.81 | 0.66      | 78.34  | 64.42  | 56.85  | 0.0195 |
> | LaMa + SDinpaint  | 51.33| 0.45 | 0.75      | 72.29  | 57.61  | 50.01  | 0.0117 |
> | SD-Inpaint + CLIPAway      | 54.93| 0.48 | 0.80      | 82.36  | 71.68  | 63.28  | 0.0095 |
>
> Incorporating SD-Inpaint as a post-process significantly improved sample quality, as indicated by the improvement in the FID score. However, the object removal metrics reveal a decline in the model’s ability to effectively remove objects. We attribute this decline to SD-Inpaint’s tendency to insert and hallucinate objects, as noted in our paper.  We also would like to mention that in KID, which is the other visual quality metric, CLIPAway model achieves better scores in addition to the significantly better scores on the object removal CLIP based metrics. We appreciate the reviewer’s suggestion, which helped us highlight the trade-offs involved in using SD-Inpaint in combination with LaMa. We will include these results in our manuscript to provide a more comprehensive evaluation of our pipeline.
>
> We also add visual results of this comparison in the attached Rebuttal Fig. 6. It can be seen that LaMa fills the erased area in a blurred way, and the SD-Inpaint, followed by LaMa, adds objects again. For example, in the first row, a horse is added to the image.
>
> **Analysis of Failure Cases**\
> We appreciate the reviewer's suggestion to present and analyze more failure cases to highlight the limitations of our method.
>
> One significant limitation of our method is the constraint on applicable image resolution as noted by the Reviewer, which is inherent to diffusion-based inpainting methods. Specifically, the model in a Stable Diffusion (SD) pipeline expects input images and masks to be in the 512x512 pixel resolution, while SD-XL pipelines expect 1024x1024 resolution. Deviating from these resolutions degrades the performance of the UNet in the diffusion pipeline, resulting in less effective inpainting, regardless of the quality of the embeddings provided by the CLIPAway pipeline. We will include a detailed discussion of these limitations in our manuscript to provide a comprehensive evaluation of our method.
>
> **Potential Extensions to Other Frameworks**\
> We appreciate the reviewer's suggestion to apply our method to SDXL. After examining the SDXL-Inpaint model, we observed that the issue of object addition is even more pronounced. SDXL recommends setting the strength parameter to 0.99, using the unerased image as input, and introducing noise at this strength (for reference: https://huggingface.co/diffusers/stable-diffusion-xl-1.0-inpainting-0.1). The strength parameter determines how much the masked portion of the reference image is transformed, ranging from 0 to 1. A higher strength adds more noise to the image, requiring more denoising steps. When the strength is set to 1, maximum noise is added, leading to degradation in image quality as noted on the webpage's limitations, with ongoing work to address this issue.
>
> Fig. 7 of the Rebuttal PDF illustrates that using the original input image with a strength of 1.0 results in lower output quality. Conversely, a strength of 0.99 often causes the model to regenerate the erased foreground object, as it retains some information about it. This setting yields a CLIP-1 score of 67.22, the lowest among those reported in Table 1 of the main paper. When combined with our CLIPAway method, the SDXL-Inpaint model achieves a CLIP-1 score of 85.91.
>
> We will include comprehensive experiments with SDXL in the revised manuscript. We believe that eliminating the reliance on unerased images will improve results for both SDXL-Inpaint and our approach.
>
> Lastly, we thank the reviewer for recommending that we open-source our code and for finding our work beneficial. We are pleased to announce that the code and model are ready and will be released soon.

---

> ### Comment · Reviewer_GcHN · 2024-08-10
> **Thanks for the rebuttal**
>
> I have carefully reviewed the authors' rebuttal and the other reviewers' comments. The authors have adequately addressed my initial concerns. I would recommend adding the additional baseline comparisons and SDXL experiments in the final version.

---

> ### Author Response · Authors · 2024-08-11
> **Thank you.**
>
> We would like to thank the reviewer for their thorough review and constructive feedback. We will certainly incorporate the additional baseline comparisons and SDXL experiments into the final version of the paper as recommended.

---

### Official Review · Reviewer_FKYu · 2024-07-12

**Soundness:** 3
**Presentation:** 2
**Contribution:** 3
**Rating:** 7
**Confidence:** 4

**Summary:**

The paper identifies a common limitation in recent diffusion model-based inpainting methods: unintended hallucinations of the removed object. To address the problem, the paper introduces CLIPAway, a plug-and-play module that does not rely on any specific training. Using vector arithmetic, it successfully obtains an embedding that predominantly focuses on the background. Through comparisons, it is demonstrated that the proposed method outperforms existing methods in object removal.

**Strengths:**

- The paper identifies a well-known limitation in the current diffusion models and the motivation behind this paper is very strong.
- The proposed method does not rely on synthetic data or any special annotations during its training, thus having a stronger ability to generalize and more robust performance compared to prior methods (Fig 8).
- The idea of using AlphaCLIP encoder and doing vector subtraction is simple but effective.
- Extensive results (Tab 1 and user study) are presented to support the conclusions of the paper, especially on whether the object is correctly removed.

**Weaknesses:**

- Although the proposed model can effectively remove the object, the quality of the inpainted image is not stable. There are notable artifacts in Fig 1, such as the shadow in the 1st, 2nd rows on the right, and 3rd, 4th rows on the left.
- The conclusions / observations in Fig 4 lacks enough evidence. E.g., in the bottom-left, it's hard to tell whether the foreground or the background is more prominent. Adding more visual results / quantitative comparisons may help to obtain a more solid/reasonable observation.
- There have been a lot of evaluations on whether the object is correctly removed (in Tab 1), but should be more on assessing the visual quality, e.g., LPIPS.

**Questions:**

- In lines 192-193, when setting up the baselines, the authors used empty prompt or "background" as input. In practice, providing an accurate prompt describing the background can also be useful in removing the object. E.g., when removing a laptop from a table, the prompt can be "an empty table". Does this work for the DM-based baselines?

**Limitations:**

Already addressed.

---

> ### Author Rebuttal · Authors · 2024-08-06
>
> We would like to thank the reviewer for their detailed feedback.
>
> **Eliminating Shadows**\
> We appreciate the reviewer’s concern about shadows in the inpainted images. Since our method uses SD-Inpaint as its backbone, inpainting is performed only on the masked regions, which ensures that only the areas the user wants to remove are altered, while the unmasked areas are preserved. LaMa, MAT, and ZITS++ follow a similar approach. To address the issue of shadows, we conducted additional experiments where both the objects and their shadows were included in the masks. As shown in the Rebuttal PDF (Fig. 3), this adjustment significantly improves the results by effectively removing both the objects and their shadows, leading to cleaner and more accurate inpainted images. This way, only the areas the user wants to remove are altered, the user can choose to keep the shadow for some artistic designs or choose to remove it.
>
> **Additional Visual Evidence**\
> We thank the reviewer for their suggestion to provide more visual evidence. We agree that additional examples would better demonstrate the efficacy of our method. To address this, we have included more visual results in the Rebuttal PDF (Fig. 2) that show the behavior of foreground-background, and projected embeddings.  Specifically, we observe that the foreground embedding is dominated by the foreground object, while the foreground object appears at a smaller scale in the conditional generation results of the background embedding. These new examples highlight the effectiveness of our projection block in removing objects of interest. We will update our submission to include these new results, strengthening our observations and making them more convincing.
>
> **Improving Visual Quality Assessment**\
> To provide a more comprehensive assessment of visual quality, we have included Kernel Inception Distance (KID) as an additional photorealism metric as given below (lower is better):
>
> | Method                        | KID    |
> |----------------------------|------|
> | ZITS                          | 0.0208 |
> | MAT                          | 0.0278 |
> | LaMa                        | 0.0195 |
> | Blended                   | 0.0362 |
> | Blended + CLIPAway          | 0.0194 |
> | Unipaint                    | 0.0360 |
> | Unipaint + CLIPAway         | 0.0199 |
> | SDInpaint                     | 0.0145 |
> | SDInpaint + CLIPAway | 0.0108 |
>
> As shown in the table, KID results align with the photorealism metrics reported previously. We did not use LPIPS because it necessitates reference images with the objects removed, which are not available. With the addition of KID, we now have three metrics to evaluate visual quality, complementing FID and CMMD, which were previously reported in the quantitative comparisons in the main paper.
>
> **Prompt Use in Baseline Comparisons**\
> We understand that using a specific prompt, such as "an empty table" when removing a laptop, can sometimes improve results. However, we found that this approach does not consistently improve the baselines.  Sample results demonstrating this are included in the Rebuttal PDF (Fig. 4).  For example, "empty table top" prompt results in inpainting the region with adding an additional component on top of the table rather than completing it directly in the first row, "an empty runway" still adds planes to the image, or "an empty bed" adds a pillow to the bed.  Additionally, generating specific prompts for every situation in a large dataset is impractical and time-consuming. Our method circumvents this issue by using vector arithmetic in the CLIP space, eliminating the need for custom prompts. We believe this makes our method more robust and user-friendly. We hope the reviewer finds this perspective agreeable.

---

> ### Comment · Reviewer_FKYu · 2024-08-13
>
> Thank you for providing extensive additional results in the rebuttal, they do improve my understanding of the proposed method and fully resolve my concerns. I raised my score, and hope the additional results can be included in the paper.

---

> > ### Author Response · Authors · 2024-08-13
> >
> > We would like to thank the reviewer for their constructive review and for raising their score based on the additional results provided in the rebuttal. We're glad to hear that the new information has addressed the reviewers concerns. We will certainly include these additional results in the final version of the paper.

---

### Official Review · Reviewer_hrfv · 2024-07-12

**Soundness:** 3
**Presentation:** 3
**Contribution:** 3
**Rating:** 5
**Confidence:** 3

**Summary:**

This paper proposes an approach that aims to tackle object removal in stable diffusion models.  The paper utilizes AlphaCLIP embedding (that are trained with an additional alpha layer mask to enable incorporation of regions of interest), and techniques such as IP-Adapter that decouples the diffusion unet cross-attention mechanisms to accept features such as coming from image/text separately. The authors train using image prompts as inputs and controlling the alpha channel to correspond to background or foreground (inverse of background) focus to guide the attention mechanism. Eq 2 aims to calculate embedding that are orthogonal to the foreground embedding, resulting in embedding that are focusing on the background.

**Strengths:**

- The method leverages existing work well, and the proposed method appears to indeed remove objects in a smoother way than competitors
-It is sometimes difficult to judge from a few examples qualitatively, but it seems that the proposed method  performs relatively better in most cases presented
- ablation studies show the impact of background/foreground embedding - although in most cases it seems that background and foreground embedding do contain both. The orthogonal embedding do appear to include only information related to the background - although that does not exactly match the background of the image

**Weaknesses:**

- As with the shadows example mentioned in the limitations, the method does not seem to remove objects based on much context - i.e. region of interest is a stronger control. This may lead to undesirable or unrealistic results - so perhaps a trade-off between strict ROI adherence and context could be employed.
- The method can be plug and play after training - but the training process itself can make it specific to architectural choices
- Although mentioned in the paper, this work appears to have a larger computational overhead that simpler inpainting methods, which perhaps can then be improved with simpler post-processing
- the paper is focused on object removal in a background-consistent manner, which is a limited task, while some compared methods tackle the more general problem if image inpainting

**Questions:**

please see above

**Limitations:**

yes

---

> ### Author Rebuttal · Authors · 2024-08-06
>
> We would like to thank the reviewer for their detailed feedback.
>
> We appreciate the reviewer's recognition of our method's ability to leverage existing work effectively and its superior performance in removing objects smoothly compared to competitors. To address the concern regarding the qualitative judgment based on a few examples, we will extend our supplementary material with additional qualitative examples to better demonstrate the effectiveness of our method.  Additionally, we would like to emphasize that the quantitative analysis provided in Table 1 shows we achieve consistent improvements over many baselines.
>
> We appreciate the reviewer's comments on our ablation studies showing the impact of background and foreground embeddings. While background and foreground embeddings may sometimes contain overlapping information, our method aims to orthogonalize these embeddings to effectively isolate relevant background information. The orthogonal embeddings are designed to include only information related to the background. Although these embeddings are not expected to exactly match the original background of the image, they are constructed to exclude the foreground object and provide a contextually coherent background representation. To further clarify and validate the effectiveness of our approach, we provide additional examples in the Rebuttal PDF (Fig. 2) which show the behavior of foreground-background, and projected embeddings. We will add more examples in the revised paper. We would like to thank the reviewer for bringing this to our attention.
>
> **Regarding Contextual Understanding**\
> Since we use SDInpaint as our backbone, inpainting is carried out only on the regions identified by the provided masks as happens in other competing methods as well, LAMA, MAT, ZITS++. To address the reviewer's concern about the method not removing shadows, we conducted new experiments where both objects of interest and their shadows were included in the corresponding masks.  We share these results in the Rebuttal PDF (Fig. 3). Our method produces highly satisfactory results, effectively eliminating the objects and their shadows if the user prefers to.
>
> **Clarification on "Plug and Play" Capability**\
> We apologize if the use of the term "plug and play" caused any confusion. Our method bridges the AlphaCLIP embedding space to the CLIP embedding space through projections, which requires some architectural decisions during the training process. However, once trained, our CLIPAway module can be integrated into any framework that employs the CLIP embedding space or other CLIP embedding spaces with similar properties — without requiring additional training. As demonstrated in Table 1, adding CLIPAway to SD-Inpaint, Blended Diffusion, and Unipaint frameworks significantly improves their performance on the object removal task.
>
> **Computational Overhead and Efficiency**\
> While diffusion models do have higher computational overhead compared to GAN models, they are currently favored due to the significant improvements they offer. Additionally, there is extensive research focused on optimizing diffusion models and reducing the number of required denoising steps to improve their efficiency. Therefore, we believe it is important to continue improving these models for various tasks. . We are not aware of any straightforward post-processing technique that can elevate simpler inpainting methods to a state-of-the-art level of quality. Reviewer GcHN brought the LaMa + SD-Inpaint baseline to our attention, where the faster LaMa model generates the initial inpainting results and SDInpaint, which is a diffusion-based inpainting method, serves as post-processing. However, this combination is more computationally intensive than our approach and yields worse results.
>
> **Expanding Beyond Object Removal**\
> While our main focus in the paper is on object removal, our framework, can be applied to other tasks. Based on the Reviewer's comment, we investigated the capabilities of our framework  on reference-based background inpainting as given in the Rebuttal PDF (Fig. 5). By computing the projected embedding of a reference image representing the target background, we inpaint the background of the input image while preserving its foreground.  We will revise our submission to include these new results, demonstrating that our method can open directions for image manipulation and general image inpainting problems. Additionally, we would like to emphasize that object removal is a significant task in its own right, and previous research has proposed generating datasets specifically for this purpose.

---

> ### Comment · Reviewer_hrfv · 2024-08-12
> **thanks for the detailed response**
>
> many thanks to the authors for the detailed response. In my view, this is still a limited task that can be dealt with in different ways with varying success. However, I am happy with the answers that the reviewers have provided and have raised score accordingly

---

> > ### Author Response · Authors · 2024-08-12
> >
> > We would like to thank the reviewer for their feedback and for raising their score. We appreciate the reviewer's acknowledgment of our detailed response and understand their perspective on the task's limitations.

---

### Official Review · Reviewer_6DqA · 2024-07-12

**Soundness:** 3
**Presentation:** 4
**Contribution:** 3
**Rating:** 7
**Confidence:** 4

**Summary:**

This paper addresses a commonly observed problem when using pre-trained diffusion models for object removal: in standard inpainting setups, these models often add similar objects in place of the ones to be removed instead of extending the background to the masked area. To address this problem, the authors use CLIPAway, a method that leverages region-focused embeddings from Alpha-CLIP, obtaining further disentangled background embeddings that can then be used to guide the diffusion model to perform high-quality object removal without foreground leakage.

**Strengths:**

- This paper addresses a very common and important problem in practical applications of large-scale pretrained T2I diffusion models.
- While very simple, the proposed method combines existing methods (Alpha-CLIP and T2I-Adapter/T2I diffusion models conditioned on CLIP embeddings) in a novel and very effective manner. The proposed method is demonstrated to work well qualitatively and outperform other approaches when combined with off-the-shelf inpainting methods in a quantiative evaluation.
- The paper is written clearly and each part of the method is well-motivated and explained both in text and with qualitative examples.

**Weaknesses:**

The main weakness with the paper is the simplicity of the method. Generally, a simple method that achieves the goal is often better than an over-involved complex method, but most of the parts of this method are relatively obvious (translating from one CLIP embedding space to another seems to primarily be an engineering decision as the pre-trained IP-Adapter does not directly consume Alpha-CLIP embeddings, but it would likely be very simple to just train an adapter on these embeddings; and orthogonalization is a very basic operation, even if it is curious that it works so well in this case) and aspects of transferability of this method, which are relevant for its further impact beyond the implementation presented in this paper (e.g., whether it is reliant on the exact two CLIP embedding spaces used, or whether they can be substituted with other CLIP spaces with similar properties), are not covered.

**Questions:**

- Can the projection also be performed in the AlphaCLIP embedding space (or other CLIP embedding spaces) or does it only work in the adapted OpenCLIP embedding space? This would be especially interesting as it would speak to the generalizability of the proposed method, as being limited to specific CLIP embedding spaces might limit practical applications, such as applying this to a standard unCLIP model (e.g., Karlo https://github.com/kakaobrain/karlo, Stable unCLIP https://huggingface.co/docs/diffusers/en/api/pipelines/stable_unclip).
- What is the time (wall clock/GPU-hours) required for training the MLP?

**Limitations:**

The authors have adequately addressed the limitations and societal impact of their work.

---

> ### Author Rebuttal · Authors · 2024-08-06
>
> We would like to thank the reviewer for their detailed review.
>
> We appreciate the reviewer's feedback regarding the simplicity of our method. We share the same preference for simplicity over complex solutions. While our solution might seem obvious now, previous research attempted to address object removal tasks by generating synthetic datasets. For instance, InstInpaint [30], MagicBrush [33], InstructPix2Pix [2], and concurrently ObjectDrop [26] explored different approaches to create training datasets for this purpose. Our method did not require a dataset generation and achieve better results than competing methods which makes it more elegant.
>
> The reviewer asks about the flexibility of CLIP embedding spaces. This is an excellent point.  Our method is not confined to the CLIP embedding spaces used in our initial experiments. To demonstrate this flexibility, we trained the MLP for AlphaCLIP with model keys VIT-L/14@336px, and VIT-B/16 in addition to the VIT-L/14 that was presented in the paper. After training, we evaluated each model using the same setup as in Table 1 of the manuscript. The qualitative scores for each model are provided in the table below:
>
> | Method | FID  | CMMD | CLIP Dist | CLIP@1 | CLIP@3 | CLIP@5 |
> |-------------------------------------------|------|------|-----------|--------|--------|--------|
> | SD-Inpaint  | 59.21| 0.54 | 0.75 | 70.45  | 57.14  | 49.88  |
> | SD-Inpaint + CLIPAway (VIT L/14 - presented in the paper) | 57.32| 0.53 | 0.81      | 84.82  | 74.42  | 67.76  |
> | SD-Inpaint + CLIPAway (VIT L/14@336px)  | 54.93| 0.48 | 0.80      | 82.36  | 71.68  | 63.28  |
> | SD-Inpaint + CLIPAway (ViT-B/16)  | 55.31| 0.48 | 0.78      | 83.57  | 72.44  | 63.81  |
>
> The results consistently improve the SD-Inpaint model with different CLIP embeddings.
>
> The reviewer asks if the projection could be done on the AlphaCLIP space rather than OpenCLIP embedding space. Our projection method is not restricted to the OpenCLIP embedding space. Since AlphaCLIP’s vision transformer is trained with similar objectives as the CLIP vision transformer, their feature spaces are conceptually similar. Therefore, projections can be performed in the AlphaCLIP embedding space or other CLIP embedding spaces with similar properties. To support this, we evaluated the projection method on the AlphaCLIP feature space (projection on AlphaCLIP space followed by MLP) using the same setup as described in Table 1 of the manuscript. The results, shown below, confirm that our projection approach is applicable beyond the OpenCLIP embedding space.
>
> | Model | FID  | CMMD | CLIP Dist | CLIP@1 | CLIP@3 | CLIP@5 |
> |------------------|------|------|-----------|--------|--------|--------|
> | SD-Inpaint  | 59.21| 0.54 | 0.75 | 70.45  | 57.14  | 49.88  |
> | SDInpaint + CLIPAway VIT-L/14 @336px | 54.46| 0.36 | 0.82 | 82.13  | 70.02  | 63.28  |
> |SDInpaint + CLIPAway   VIT-L/14 | 56.15| 0.42 | 0.86 | 85.31  | 74.26  | 68.58  |
> | SDInpaint + CLIPAway  ViT-B/16 | 54.99| 0.41 | 0.84 | 85.08  | 74.79  | 68.35  |
>
> We will include these results in the revised paper. We believe they will make the paper more comprehensive. We would like to thank the reviewer for these valuable suggestions.
>
> We also provided qualitative example object removal results in the Rebuttal PDF (Fig. 1), further confirming the effectiveness of our projection approach across different CLIP embedding spaces. These results illustrate our method’s flexibility and robustness, supporting its broader applicability.
>
> Training the MLP layer for AlphaCLIP with the model key VIT-B/16 using a single Nvidia A40 GPU takes approximately 7 hours which is a negligible GPU time compared to training the diffusion models from scratch.

---

> > ### Comment · Reviewer_6DqA · 2024-08-12
> >
> > Thank you for the extensive response and for running the extensive additional experiments in the short rebuttal timespan!
> >
> > I agree that the method presented is very elegant due to its simplicity. My main concern was that it was not clear from the initial submission whether this simplicity would limit it to the specific model combination presented in the paper. Given the extensive additional results demonstrating that the method works across different AlphaCLIP versions, the projection can be performed in the AlphaCLIP space as well, and the additional results on SDXL demonstrating that the performance is not limited to a specific diffusion model, this concern has been thoroughly addressed.
> >
> > After carefully reviewing the authors' rebuttal and responses to mine and the other reviews, I will raise my score from 6 to 7 and hope for acceptance. I'd appreciate it if the authors incorporated these additional results into the paper in a suitable manner.

---

> > > ### Author Response · Authors · 2024-08-12
> > >
> > > We would like to thank the reviewer for their feedback which helped us to improve the paper. We also would like to thank the reviewer for taking the time to review the additional results and responses, and we appreciate the reviewer raising their score. We will certainly incorporate the additional results into the paper.

---

### Author Rebuttal · Authors · 2024-08-07

We want to thank all reviewers for their valuable feedback. We have responded to each reviewer's questions in the rebuttal sections, and attached a PDF file with figures for our additional results.

---

### Decision · Program_Chairs · 2024-09-25

**Decision:**

Accept (poster)

**Comment:**

After rebuttal, all the reviewers unanimously vote for acceptance of this work. After checking the rebuttal, the review, and the manuscript, the AC recommends acceptance.